# Zero-Shot-BERT-Adapters: a Zero-Shot Pipeline for Unknown Intent Detection

**Daniele Comi**
IBM Consulting Italy
daniele.comi@ibm.com

**Dimitrios Christofidellis**
IBM Research Europe
dic@zurich.ibm.com

**Pier Francesco Piazza**
IBM Consulting Italy
pier.francesco.piazza@it.ibm.com

**Matteo Manica**
IBM Research Europe
tte@zurich.ibm.com

## Abstract

Intent discovery is a crucial task in natural language processing, and it is increasingly relevant for various of industrial applications. Identifying novel, unseen intents from user inputs remains one of the biggest challenges in this field. Herein, we propose Zero-Shot-BERT-Adapters, a two-stage method for multilingual intent discovery relying on a Transformer architecture fine-tuned with Adapters. We train the model for Natural Language Inference (NLI) and later perform unknown intent classification in a zero-shot setting for multiple languages. In our evaluation, we first analyze the quality of the model after adaptive fine-tuning on known classes. Secondly, we evaluate its performance in casting intent classification as an NLI task. Lastly, we test the zero-shot performance of the model on unseen classes, showing how Zero-Shot-BERT-Adapters can effectively perform intent discovery by generating semantically similar intents, if not equal, to the ground-truth ones. Our experiments show how Zero-Shot-BERT-Adapters outperforms various baselines in two zero-shot settings: known intent classification and unseen intent discovery. The proposed pipeline holds the potential for broad application in customer care. It enables automated dynamic triage using a lightweight model that can be easily deployed and scaled in various business scenarios, unlike large language models. Zero-Shot-BERT-Adapters represents an innovative multi-language approach for intent discovery, enabling the online generation of novel intents. A Python package implementing the pipeline and the new datasets we compiled are available at the following link: https://github.com/GT4SD/zero-shot-bert-adapters.

## 1 Introduction

Language Models (LM) and, in general, Natural Language Processing (NLP) methodologies have a pivotal role in modern communication systems.

In dialogue systems, understanding the actual intention behind a conversation initiated by a user is fundamental, as well as identifying the user's intent underlying each dialogue utterance (Ni et al., 2021). Recently, there has been a growing interest in such applications, evidenced by the plethora of datasets and competitions established to tackle this problem. (Science, 2022; Casanueva et al., 2020; Zhang et al., 2021). Unfortunately, most of the available most of the available production systems leverage models trained on a finite set of possible intents, with limited or no possibility to generalize to novel unseen intents (Qi et al., 2020). Such limitations constitute a significant blocker for most applications that do not rely on a finite set of predetermined intents. They, instead, require a constant re-evaluation based on users' inputs and feedback. Additionally, most of the datasets for intent classification are mainly available for the English language, significantly limiting the development of dialogue systems in other languages. Expanding the set of available intents online is imperative to build better dialogue systems closely fitting users' needs. Besides better-matching user expectations, this also enables the definition of new common intents that can slowly emerge from a pool of different users. This is a key aspect of building systems that are more resilient over time and can follow trends appropriately. Current supervised techniques fall short of tackling the challenge above since they cannot usually discover novel intents. These models are trained by considering a finite set of classes and cannot generalize to real-world applications (Larson et al., 2019). Here, we propose a multi-language system that combines dependency parsing (Honnibal and Johnson, 2015; Nivre and Nilsson, 2005; ExplosionAI, 2015) to extract potential intents from a single utterance and a zero-shot classification approach. We rely on a BERT Transformer backbone (Vaswani et al., 2017; Devlin et al., 2018) fine-tuned for NLI (Nat-

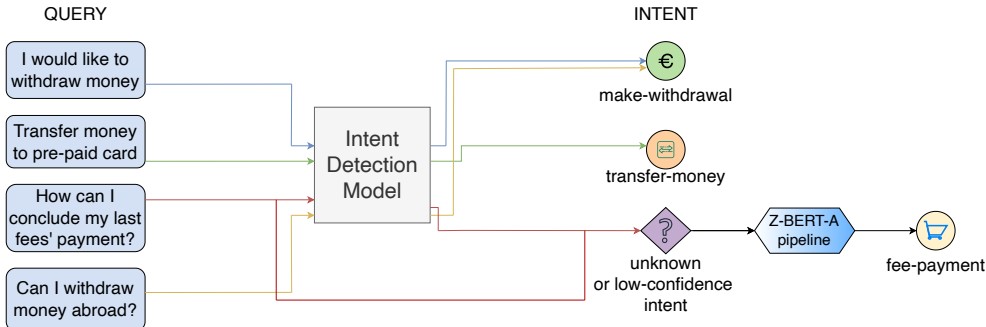

Figure 1: Deployment scenario, where the current intent detection (classification) model is initially used to examine user input and based on the classification outcome problematic cases are forwarded to Zero-Shot-BERT-Adapters for intent discovery.

ural Language Inference) (Xia et al., 2018; Yin et al., 2019) to select the intent that is best fitting the utterance in a zero-shot setting. In the NLI fine-tuning, we leverage Adapters (Houlsby et al., 2019; Pfeiffer et al., 2020) to reduce memory and time requirements significantly, keeping base model parameters frozen. We evaluated our approach, Zero-Shot-BERT-Adapters, for English, Italian, and German, demonstrating the possibility of its application in both high and low-resource language settings. The design focuses on a production setting where automatic intent discovery is fundamental, e.g., customer care chatbots, to ensure smooth user interaction. Figure 1 depicts how the Zero-Shot-BERT-Adapters pipeline fits in an intent detection system. From now on, for simplicity, we will refer to Zero-Shot-BERT-Adapters as Z-BERT-A in Tables and Figures.

## 2 Related Literature

There have been various efforts aiming at finding novel and unseen intents. The most popular approach is a two-step method where, initially, a binary classifier determines whether an utterance fits the existing set of intents, and, finally, zero-shot techniques are applied to determine the new intent (Xia et al., 2018; Siddique et al., 2021; Yan et al., 2020). (Liu et al., 2022) proposed an approach leveraging Transformers-based architectures, while most relied on RNN architectures like LSTMs (Xia et al., 2018). For our problem of interest, two notable efforts have attempted to address the issue of unseen intent detection (Vedula et al., 2019; Liu et al., 2021). (Liu et al., 2021) handled novel intent discovery as a clustering problem, proposing an adaptation of K-means. Here, the authors leverage dependency parsing to extract from

each cluster a mean ACTION-OBJECT pair representing the common emerging intent for that particular cluster. Recently, Large Language Models (LLMs) achieved remarkable zero-shot generalization (Chowdhery et al., 2022; Sanh et al., 2021; Parikh et al., 2023) capabilities. (Parikh et al., 2023) report benchmarks on intent classification using GPT-3 (Chowdhery et al., 2022) and analyze performance both in zero and few-shot settings. While these solutions are compelling, they are only an ideal fit for some real-world use cases. Models of these sizes can be expensive to operationalize, especially in on-premise settings with strict hardware constraints. While there is no literature on the novel intent discovery and zero-shot classification in a multilingual setting, there has been a recent attempt to perform zero-shot intent detection (Liu et al., 2019). The authors study the problem of intent detection using attention-based models and cover low-resource languages such as Italian. Other approaches leveraging multilingual solutions, such as Nicosia et al., focus on a zero-shot multilingual semantic parsing task. Here, the authors present a novel method to produce training data for a multilingual semantic parser, where machine translation is applied to produce new data for a slot-filling task in zero-shot.

## 3 Background

In intent discovery, we aim at extracting from a single utterance $x$ a set of potentially novel intents $y_i$ and automatically determine the best fitting one for the considered utterance. We can cast intent discovery as a Natural Language Inference (NLI) problem. We can rely on a language model $\phi(x, \gamma(x))$ to predict the entailment between the utterance $(x)$ and a set of hypotheses based on the candidate

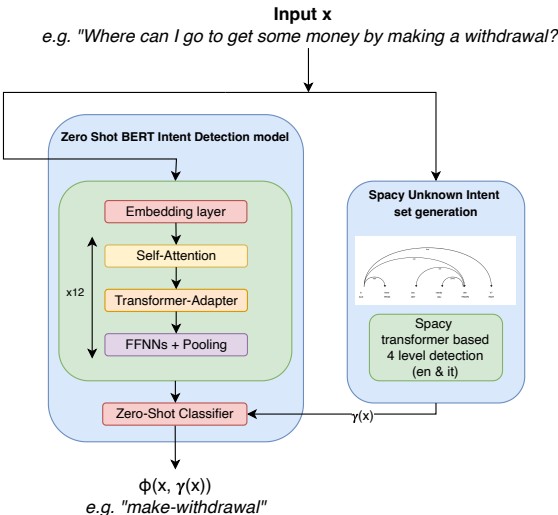

**Input x**
*e.g. "Where can I go to get some money by making a withdrawal?"*

φ(x, γ(x))
*e.g. "make-withdrawal"*

Figure 2: Zero-Shot-BERT-Adapters architecture. The two stages are shown: the first stage on the right where the intent generation algorithm 1 is run to obtain language-dependent intent candidates γ(x) which are then used as labels for the zero-shot classification in the second stage.

intents (γ(x)) where γ is a function used to extract hypotheses from the set of potential intents $y_i$. As previously shown by Xian et al., using NLI models for zero-shot classification represents a practical approach in problems where the set of candidate intents is known. In practice, the classification problem becomes an inference task where a combination of a premise and a hypothesis are associated with three possible classes: *entailment, neutral* and *contradiction*. Yin et al. have shown how this approach allows considering an input hypothesis based on an unseen class. It then generates a probability distribution describing the entailment from the input premise-hypothesis pair, hence a score correlating the input text to the novel class. While this technique is highly flexible and, in principle, can handle any association between utterance and candidate intent, determining good candidates based on the analyzed utterance remains a significant challenge.

## 4  Approach

Herein, we focus on building a pipeline to handle unseen classes at inference time. In this context, we need both to generate a set of candidate intents from the considered utterance and to classify the provided input against the new set of candidate intents. We tackle the problem by implementing a pipeline in two stages. In the first stage,

we leverage a dependency parser (Honnibal and Johnson, 2015; Nivre and Nilsson, 2005) to extract a set of potential intents by exploiting specific arc dependencies between the words in the utterance. In the second stage, we leverage the set of potential intents as candidate classes for the utterance intent classification problem using a zero-shot approach (Xian et al., 2018) based on NLI relying on a BERT-based model (Devlin et al., 2018), as depicted in Figure 2. The model is tuned with Adapters (Houlsby et al., 2019) for the NLI task (BERT-A) and is then prompted with premise-hypothesis pairs for zero-shot classification on the candidate intents, completing the Zero-Shot-BERT-Adapters pipeline. The entire pipeline's implementation follows the Hugging Face pipeline API from the `transformers` library (Wolf et al., 2019).

**Intent generation**   Defining a set of potential intents from an input utterance is crucial for effective intent discovery, especially when dealing with a multilingual context. To provide this set of potentially unseen candidates, we choose to exploit the dependency parser from spaCy (Honnibal and Johnson, 2015; Honnibal et al., 2020), considering: for English the model `en_core_web_trf`, for Italian the model `it_core_news_lg`, and, for German the model `de_dep_news_trf` (all implemented in spaCy-transformers (ExplosionAI, 2019)). We extract a pair of words from an input sentence through the dependency parser. We define each pair by searching for specific Arc-Relations (AR) in the dependency tree of the parsed sentence. Unfortunately, this approach is language-dependent, as different languages can require a specific pattern-matching for the ARs.

Since intents are usually composed by an action-object pair (Vedula et al., 2019; Liu et al., 2021), we exploit this pattern for all languages when looking for candidate arc relationships. We search for DOBJ, compound, and AMOD arc relations in English. We must adjust this in Italian and search for AUX, ADVMOD, and compound arc relations. We further adapt the tags to RE, SB, and MO in German. We perform a four-level detection, which means finding the four primary relations that can generate a base intent using these relations. Once these relations, or a subset of them, are found, we add for all the languages the pairs composed by (VERB, NOUN) and, for English only, (ADJ, PRONM) with the most out/in going arcs. We refer to Appendix A for the Tables 11, 12 and 13

for a complete definition of the AR and Part-of-Speech (POS) tags considered. We lemmatize the extracted potential intents using NLTK (Loper and Bird, 2002). Lemmatization is applied to verbs and nouns independently. The lemmatized intents represent the labels that our model uses for zero-shot classification. Algorithm 1 details in pseudocode the pipeline for the intent generation with $\Theta(n)$ as time complexity, where $n$ is the number of dependency arcs in an utterance.

**Zero-shot classification** The generated potential intents are the labels for the zero-shot BERT-based classifier implemented using NLI that scores the entailment between the utterance used as a premise and the hypothesis based on the intent. Given an input utterance, we select the intent related to the pair with the highest score. We use sentence embedding vectors to define the scores (Reimers and Gurevych, 2019).

## 5 Datasets

We consider two datasets in our analysis: SNLI (Bowman et al., 2015) and Banking77-OO (Casanueva et al., 2020; Zhang et al., 2021):

- The SNLI corpus (Bowman et al., 2015) is a collection of 570k human-written English sentence pairs manually labeled as entailment, contradiction, and neutral. It is used for natural language inference (NLI), also known as recognizing textual entailment (RTE). The dataset comes with a split: 550152 samples for training, 10000 samples for validation, and 10000 samples for testing. Each sample is composed of a premise, a hypothesis, and a corresponding label indicating whether the premise and the hypothesis represent an entailment. The label can be set to one of the following: entailment, contradiction, or neutral.

- Banking77-OOS (Casanueva et al., 2020; Zhang et al., 2021) is an intent classification dataset composed of online banking queries annotated with their corresponding intents. It provides a very fine-grained set of intents in the banking domain. It comprises 13,083 customer service queries labeled with 77 intents. It focuses on fine-grained single-domain intent detection. Of these 77 intents, Banking77-OOS includes 50 in-scope intents,

and the ID-OOS queries are built up based on 27 held-out in-scope intents.

We also explore the effects of pretraining leveraging an NLI adaptation of Banking77 (Yin et al., 2019). To investigate the impact of pretraining on similar data, we extended the Banking77 dataset by casting the intent classification task as NLI. We consider the input utterance the premise and extract the most relevant word associated with it using KeyBERT (Sharma and Li, 2019). The word is then used to generate an entailed hypothesis based on the corresponding synset definition from WordNet via NLTK (Loper and Bird, 2002; Miller, 1995). Exemplar samples of synthetic Banking77 NLI data are reported in Table 1. For the hypotheses that are not considered entailed, we repeat the procedure for randomly sampled unrelated words. This process enabled us to consider the training split of Banking77-OOS for adaptive fine-tuning of the NLI model component. We call this generated dataset Banking77-OOS-NLI.

To analyze the validity of our approach in a multi-lingual setup with lower data availability, we compiled and made available both an Italian and a German version of the Banking77 dataset. We followed a two-step process involving translation using the Google Translation API (Google, 2023) and manual annotation and correction by human agents to maintain data quality following the protocol described in (Bellomaria et al., 2019).

## 6 Training

We fine-tune two versions of BERT-A (BERT-based transformer with Adapters). The first version is trained for NLI on the SNLI dataset using the original split for training, validation, and testing (Bowman et al., 2015). The second version also considers the previously introduced Banking77-OOS-NLI which keeps the split of in-scope and the out-of-scope ratio of $50 - 27$ out of 77 total intents. Depending on the language considered we use different BERT pretrained models: for English we use bert-base-uncased (Devlin et al., 2018); for Italian we use bert-base-italian-xxl-uncased (dbmdz, 2020); for German we use bert-base-german-cased (Deepset, 2019). During training, we keep the BERT backbone frozen and only update the added Adapter layers to minimize training time and memory footprint. By freezing all the original layers and letting the model be trained only on the adaptive layers, we end up with

Table 1: Sample transformations of Banking77 dataset instances from intent classification to NLI. SD is the Synset definition extracted from WordNet, which we use to generate an augmented NLI dataset as a starting point for any Text Classification dataset. A similar set of samples for the Italian version is available in the Appendix.

| Utterance | Extracted key-phrase | Wordnet Synset definition (SD) | Generated hypothesis |
|---|---|---|---|
| where do you support? | support | the activity of providing for or maintaining by supplying with necessities | this text is about **SD** |
| card delivery? | delivery | the act of delivering or distributing something | this text is about **SD** |
| last payment? | payment | a sum of money paid or a claim discharged | this text is about **SD** |

896'066 trainable parameters. The final model then has almost 111 million parameters (110 million parameters from the BERT-base-model and 896'066 parameters added through the adapters, all with a default size of 32 bit floating point). All training runs relied on the AdamW (Loshchilov and Hutter, 2017) optimizer with a learning rate set to $2 \cdot 10^{-5}$ and a warm-up scheduler. The models have been fine-tuned for a total of 6 epochs using early stopping. Our computing infrastructure for training the models run on Nvidia Tesla T4 GPUs using PyTorch and Python 3.7. In this configuration, each training epoch took approximately one minute to run. During the testing phase each run took around 36 seconds, including the overhead due to metrics computation.

# 7 Evaluation

First, we analyzed the performance of the BERT-A component and evaluated its results on a NLI task using accuracy, precision, and recall. Afterward, we compared its result on the zero-shot classification task with other available models on the same Banking77 split using accuracy. In this initial evaluation, the intents were known. The baselines considered in this setting were BART0 (Lin et al., 2022), a multitask model with 406 million parameters based on Bart-large (Lewis et al., 2019) based on prompt training; and two flavors of Zero-Shot DDN (ZS-DNN) (Kumar et al., 2017) with both encoders Universal Sentence Encoder (USE) (Cer et al., 2018) and SBERT (Reimers and Gurevych, 2019).

In the unknown intent case, we compared the pipeline against a set of zero-shot baselines based on various pre-trained transformers. As baselines, we included bart-large-mnli (Yin et al., 2019) as it achieves good performance in zero-shot sequence classification. We used this model as an alternative to our classification method, but in this case, we maintained our dependency parsing strategy

for the intent generation. As a hypothesis for the NLI-based classification, we used the phrase: "This example is $< label >$". Recently, large LMs have demonstrated remarkable zero-shot capabilities in a plethora of tasks. Thus, we included four LLMs as baseline models, namely T0 (Sanh et al., 2021), GPT-J (Wang and Komatsuzaki, 2021), Dolly (lab, 2023) and GPT-3.5 (Brown et al., 2020) using text-davinci-003 via OpenAI API (base model behind InstructGPT and GPT-3.5 (Ouyang et al., 2022; OpenAI, 2022)). In these models, the template prompt defines the task of interest. We examined whether such LLMs can serve the end-to-end intent extraction (intent generation and classification). We investigated them both in a completely unsupervised, i.e., a zero-shot settingr and using intents generated with our dependency parsing method. In the former case, the given input is just the utterance of interest, while in the latter case, the provided input includes the utterance and the possible intents. In both cases, the generated output is considered as the extracted intent. Table 7 (in Appendix) reports all prompts in different languages used to query the LLMs. We use prompts where the models generate the intent without providing a set of possible options (prompts 1 and 2). Prompts that contain the candidate intents extracted using the first stage of our pipeline (prompts 3 and 4). Regarding GPT-3.5, we used two different prompts (prompts 5 and 6) reported in the results table. These are the two best performing prompts and they were built following the suggested examples from OpenAI. In this setting, the intents generated can't match perfectly the held-out ones. For this reason, we chose to measure the performance using a semantic similarity metric, based on the cosine-similarity between the sentence embeddings of the ground-truth intents and the generated ones (Vedula et al., 2019). To set a decision boundary, we relied on a threshold based on the distributional properties of the computed similarities. The threshold $t$ is defined in

Table 2: Accuracy of the BERT-A module on each corresponding test set for the NLI task. It shows the performance differences when using SNLI, Banking77 augmented datasets in three different languages. These are the results of five different evaluation runs, on the described conditions in the training section, together with validation performances. We are indicating with an *-IT* or *-DE* suffix results for Italian and German respectively. In those cases BERT-A indicates the Adapters fine-tuned version of the respective base models.

| Model | Dataset | Test Accuracy | Precision | Recall | F1-score | Validation Accuracy |
|---|---|---|---|---|---|---|
| BERT-A | Banking77-OOS-NLI-IT | 0.841 | 0.853 | 0.852 | 0.847 | 0.863 |
| BERT-A | Banking77-OOS-NLI-DE | 0.832 | 0.844 | 0.841 | 0.838 | 0.851 |
| BERT-A | SNLI | 0.866 | 0.871 | 0.869 | 0.870 | 0.902 |
| BERT-A | Banking77-OOS-NLI | 0.882 | 0.894 | 0.894 | 0.890 | 0.894 |

Table 3: Accuracy of the BERT-A module for the zero-shot intent classification task, on Banking77-OOS compared with the considered baselines. This test is to provide a view of how Zero-Shot-BERT-Adapters and the other baseline models are performing on the task of zero-shot classification task on the test set, without having trained the model on the classes present in the test set.

| Model | Accuracy |
|---|---|
| BART0 | 0.147 |
| ZS-DNN-USE | 0.156 |
| **BERT-A** (SNLI) | 0.204 |
| ZS-DNN-SBERT | 0.390 |
| **BERT-A** (Banking77-OOS-NLI-DE) | 0.396 |
| **BERT-A** (Banking77-OOS-NLI-IT) | 0.404 |
| **BERT-A** (Banking77-OOS-NLI) | 0.407 |

Equation 1.

$$t = \begin{cases} 0.5 & \text{if } \mu \leq 0.5 \\ \mu + \alpha \cdot \sigma^2 & \text{otherwise} \end{cases} \tag{1a}$$

$$\mu = \frac{1}{n} \sum_{i=1}^{n} \frac{x_i \cdot y_i}{||x_i|| \cdot ||y_i||} \tag{1b}$$

$$\sigma^2 = \frac{1}{n} \sum_{i=1}^{n} \left( \frac{x_i \cdot y_i}{||x_i|| \cdot ||y_i||} - \mu \right)^2 \tag{1c}$$

where $\alpha$ is an arbitrary parameter to control the variance impact, which we set to 0.5 in our study.

The evaluation of the pipeline was repeated five times to evaluate the stability of the results.

## 8 Results

Firstly, we report the performance of BERT-A on the NLI task, see Table 2. Here, BERT-A obtained the best results when being fine-tuned on Banking77-OOS-NLI. The accuracy, precision,

and recall achieved confirm the quality of the pre-trained model which achieves a 1.6% in accuracy and 2% in F1-score then BERT-A fine-tuned with SNLI. We also report the results on the Italian and German fine-tuned models to highlight the impact of the different languages.

Table 3 shows how the Z-BERT-A component improves results over most baselines in terms of accuracy in the known intent scenario where we present all available classes to the model as options in a zero-shot setting. Remarkably, the BERT-A version fine-tuned on Banking77-OOS-NLI outperforms all the considered baselines. Also, when considering at the German and Italian version of BERT-A we obseve comparable performance with the English counterpart, showing again how our approach can adapt to different scenarios.

Finally, we report Zero-Shot-BERT-Adapters on the unknown intent discovery task. Tables 4 and 5 show the performance of BERT-A fine-tuned on SNLI and Banking77-OOS-NLI in comparison with a selection of zero-shot baselines for intent discovery. For the baselines, we show the performance on zero-shot classification given potential intents from our approach (prompts 3, 4, and 6) and on new intents discovery solely using the input utterance (prompts 1, 2, and 5). Table 7 in Appendix A reports all prompts considered. Both flavors of Zero-Shot-BERT-Adapters outperform the considered baselines by a consistent margin, by 2.7%, against the best baseline of T0. In the multilingual setting, Table 5 and 6, we did not consider GPT-J and Dolly as a baselines because they partially or completely lack support for Italian and German. Our pipeline applied for the Italian language achieves remarkable results. It outperforms the best baseline, T0, by 13% in cosine similarity score. It also shows how LLMs' performance exhibit a bias towards low-resource languages. For German, we have observed again better perfomance com-

Table 4: Cosine similarity between ground-truth and generated intents for the full Zero-Shot-BERT-Adapters pipeline and the other baseline models on the ID-OOS test set of Banking77-OOS. We hereby include on the final desired results also a range in which the Cosine similarity is found on the multiple runs.

| Model | Prompt | Cosine similarity | $t$ |
|---|---|---|---|
| GPT-J | en-1 | 0.04 | 0.5 |
| GPT-J | en-2 | 0.04 | 0.5 |
| GPT-J | en-3 | 0.117 | 0.5 |
| GPT-J | en-4 | 0.098 | 0.5 |
| T0 | en-1 | 0.148 | 0.5 |
| T0 | en-2 | 0.189 | 0.5 |
| T0 | en-3 | 0.465 | 0.5 |
| T0 | en-4 | 0.446 | 0.5 |
| Dolly | en-1 | 0.316 | 0.5 |
| Dolly | en-2 | 0.351 | 0.5 |
| Dolly | en-3 | 0.384 | 0.5 |
| Dolly | en-4 | 0.333 | 0.5 |
| GPT-3.5 | en-5 | 0.326 | 0.5 |
| GPT-3.5 | en-6 | 0.421 | 0.5 |
| bart-large-mnli | - | 0.436 | 0.5 |
| **Z-BERT-A** (SNLI) | - | **0.478** $\pm$ 0.003 | 0.5 |
| **Z-BERT-A** (Banking77-OOS-NLI) | - | **0.492** $\pm$ 0.004 | 0.546 $\pm$ 0.011 |

Table 5: Cosine similarity between ground-truth and generated intents for the full Zero-Shot-BERT-Adapters pipeline and the other baseline models on the ID-OOS set of Italian Banking77-OOS. We hereby include on the final desired results also a range in which the Cosine similarity is found on the multiple runs.

| Model | Prompt | Cosine similarity | $t$ |
|---|---|---|---|
| T0 | it-1 | 0.127 | 0.5 |
| T0 | it-2 | 0.145 | 0.5 |
| T0 | it-3 | 0.337 | 0.5 |
| T0 | it-4 | 0.302 | 0.5 |
| GPT-3.5 | it-5 | 0.24 | 0.5 |
| GPT-3.5 | it-6 | 0.266 | 0.5 |
| **Z-BERT-A** | - | **0.467** $\pm$ 0.002 | 0.5 |

Table 6: Cosine similarity between ground-truth and generated intents for the full Zero-Shot-BERT-Adapters pipeline and the other baseline models on the ID-OOS set of German Banking77-OOS. We hereby include on the final desired results also a range in which the Cosine similarity is found on the multiple runs.

| Model | Prompt | Cosine similarity | $t$ |
|---|---|---|---|
| T0 | de-1 | 0.158 | 0.5 |
| T0 | de-2 | 0.192 | 0.5 |
| T0 | de-3 | 0.321 | 0.5 |
| T0 | de-4 | 0.318 | 0.5 |
| GPT-3.5 | de-5 | 0.05 | 0.5 |
| GPT-3.5 | de-6 | 0.17 | 0.5 |
| **Z-BERT-A** | - | **0.36** $\pm$ 0.007 | 0.5 |

pared to the baselines, even though, in this comparison, T0 is competitive with our method. It is interesting to observe how approaches relying on smaller models like Zero-Shot-BERT-Adapters can outperform LLMs. The results show that combining smaller models with focused simpler classical NLP approaches can still bring better results in intent discovery and detection. The baselines consisting of LLMs, while achieving acceptable results, are not able to get a similarity score as high as Zero-Shot-BERT-Adapters. This indicates that lightweight approaches that can be easily deployed and scaled are a viable sollution for task-focused scenarios.

Figure 3 reports the average cosine similarity be-

tween the generated intents for each of the ground-truth intents in the multilingual setting. It combines the results between the three different language settings. We can see how for various corresponding intents, Zero-Shot-BERT-Adapters is achieving proportionally similar results in all the languages. As expected, for some classes the results on the English language settings are superior.

## 9 Conclusions and Future Work

We proposed a pipeline for zero-shot prediction of unseen intents from utterances. We performed a two-fold evaluation. First, we showed how our BERT-based model fine-tuned with Adapters on

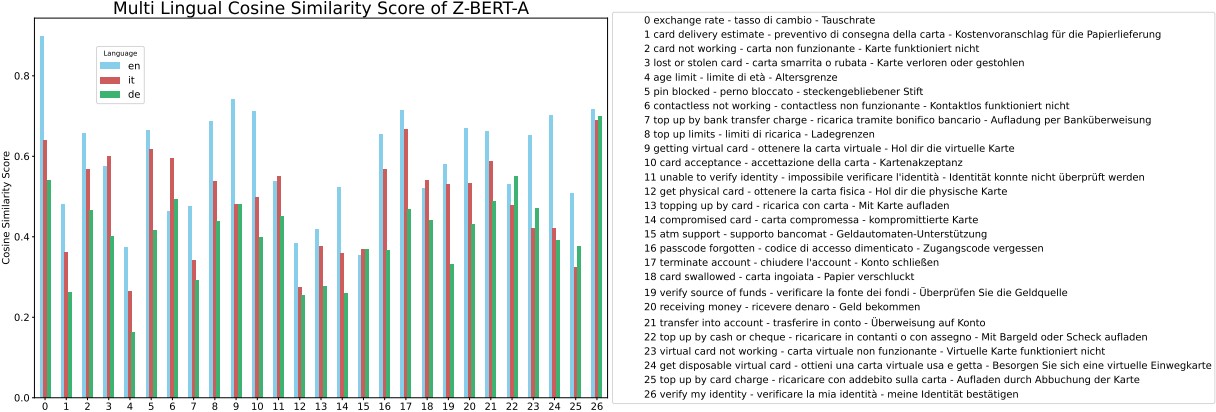

Figure 3: Bar plot showing the average cosine-similarity for each unseen-intent class in the multilingual settings. The corresponding numerical classes for English, Italian and German are also displayed as legend on the right.

NLI can outperform a selection of baselines on the prediction of known intents in a zero-shot setting. Secondly, we evaluated the full pipeline capabilities comparing its performance with the results obtained by prompting LLMs in an unknown intent set considering multiple languages. These results prove that our solution represents an effective option to extend intent classification systems to handle unseen intents, a key aspect of modern dialogue systems for triage. Moreover, using a relatively lightweight base model and relying on adaptive fine-tuning, the proposed solution can be deployed in the context of limited resources scenarios, e.g., on-premise solutions or small cloud instances. In this way, we don't compromise on model performance, as we showed that smaller models like the ones powering our pipeline can outperform LLMs in intent discovery. An interesting avenue to explore in the future consists in relying on zero-shot learning approaches in the intent generation phase (Liu et al., 2021) without compromising on model size and inference requirements. Zero-Shot-BERT-Adapters and the Italian and German datasets are available at the following link: https://github.com/GT4SD/zero-shot-bert-adapters.

## 10 Ethical considerations

Dialog systems are a helpful tool that companies and organizations leverage to respond to inquiries or resolve issues of their customers promptly. Unfortunately, an uncountable number of users interact with such tools on various occasions, which inherently poses bias concerns. We believe that using Zero-Shot-BERT-Adapters in intent detection

pipelines can help mitigate bias by dynamically adapting intents based on the user audience interacting with the system. Relying on the new emerging intents from Zero-Shot-BERT-Adapters, one can easily compile a dataset to fine-tune an existing intent detection model that carries an inherent bias coming from its training dataset. Moreover, our approach aims to minimize the usage of computing resources by relying on a relatively small model that requires minimal resources for training and inference. This parameter efficiency is relevant when considering the carbon footprint associated with training and serving of recent LLMs.

## 11 Limitations

The main limitation of this pipeline currently lies in the new intent generation stage where we are using classic dependency parsing to generate potential new intents. This is a limitation for two main reasons. Firstly, because we are bound by the input utterance when producing an intent, risking to lack in terms of generalization power. Secondly, when multiple languages are involved the result of the dependency parser has to be interpreted in a language-dependent way. As shown in Tables 11 and 12, different languages call for different arc-relationships. This limitation is something that can be addressed by replacing the dependency parsing with a model-based zero-shot approach that would be the natural extension of the presented method.

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

## A Intent generation and results

---

**Algorithm 1** Intent generation algorithm

---

**Input**: utterance $x$, language $l$
**Output**: set of potential intents $I$

1: let deps = depedency_parser($x$)
2: let ARs = {'en': ['DOBJ', 'AMOD', 'compound'], 'it': ['AUX', 'ADVMOD', 'compound']}
3: **for** arc in deps **do**
4:   **if** arc['label'] in ARs[$l$] **then**
5:     Let start = beginning word of the arc relationship
6:     Let end = ending word of the arc relationship
7:     $I$.append(start['word'] + end['word'])
8:   **end if**
9: **end for**
10: Let best_w = set of words with most in/outgoing arcs
11: **for** (word_1, word_2) in best_w as (VERB, NOUN) or (ADJ, PRONM) **do**
12:   $I$.append(word_1 + word_2)
13: **end for**
14: $I = \{$lemmatize($i$) for $i$ in $I\}$
15: **return** $I$

---

Tables 11, 12 and 13 report the POS tags for the arc-relationship chosen for the different language settings. These tags are used for choosing the best zero-shot class candidates through the dependency parser obtained from the input utterance.

Algorithm 1 describes in detail our intent generation algorithms. It relies on the Spacy Transformer Dependency Parser on which we apply the previous explained AR and POS language-dependent relations. The algorithm takes in account information regarding this AR language-dependent relations. It is then able to find various potential intents from an input utterance. We also show the testing prompts table 7, where each prompt used for testing is shown. For both Italian and English the prompts are the same, they are just correctly translated and language-adapted. For the various baselines we are using four different prompts, two with potential intents in input, obtained through the pipeline of Zero-Shot-BERT-Adapters and two with only the utterance. For ChatGPT's GPT-3.5 model we used different specific prompt obtained from OpenAI suggested prompt examples.

Table 11: Arc-Relations (AR) and Part-of-Speech (POS) tags used in the intent generation phase for the English language.

| AR/POS tag | Description |
|---|---|
| VERB | verb, covering all verbs except for auxiliary verb |
| NOUN | noun, corresponds to all cases of singular or plural nouns |
| ADJ | adjective, covering also relative and superlative adjectives |
| PRONM | pronoun, all kinds of pronouns |
| DOBJ | direct object, noun phrase which is the (accusative) object of the verb |
| AMOD | adjectival modifier that serves to modify the meaning of the noun phrase |
| compound | noun compounds |

Table 12: Arc-Relations (AR) and Part-of-Speech (POS) tags used in the intent generation phase for the Italian language.

| AR/POS tag | Description |
|---|---|
| VERB | verb, covering all verbs except for auxiliary verb |
| NOUN | noun, corresponds to all cases of singular or plural nouns |
| AUX | auxiliary verbs |
| ADVMOD | adjectival modifier that serves to modify the meaning of the noun phrase |
| compound | noun compounds |

Table 13: Arc-Relations (AR) and Part-of-Speech (POS) tags used in the intent generation phase for the German language.

| AR/POS tag | Description |
|---|---|
| VERB | verb, covering all verbs except for auxiliary verb |
| NOUN | noun, corresponds to all cases of singular or plural nouns |
| AUX | auxiliary verbs |
| PROPN | A proper noun used in a sentence for a specific object. |
| compound | noun compounds |

As well as for the original Banking77 dataset we present in Table 9 some Italian exemplar samples and in Table 10 some German examplar samples. These are synthetic samples for the italian and german Banking77 NLI adapted dataset. Concluding, to appreciate the quality of the generated intents, in Table 14 we report some examples of unseen ground-truth intents and the corresponding Zero-Shot-BERT-Adapters predictions. It is interesting to see how intents obtained through the pipeline can be sometimes equal other than just semantically equal. The italian and German intents are also listed. It's interesting to see how changing the language can lead to intents, while semantically similar, different from other languages.

Table 7: Prompts used for evaluating LLMs.

| Prompt name | Prompt text |
|---|---|
| en-1 | Considering this utterance: [utterance]. What is the intent that best describes it? |
| en-2 | Considering this utterance: [utterance]. What is the intent that best describes it expressed as a phrase of one or two words? |
| en-3 | Given the utterance: [utterance]. What is the best fitting intent, if any, among the following: [potential intents]? |
| en-4 | [utterance] Choose the most suitable intent based on the above utterance. Options: [potential intents] |
| en-5 | Extract the intent based on this utterance: [utterance]. |
| en-6 | Given the following intents: [potential intents]. Choose the best fitting, if any, for this utterance: [utterance]. |
| it-1 | Considerando questa frase: [utterance]. Quale è l'intento che la descrive meglio? |
| it-2 | Considerando questa frase: [utterance]. Quale è l'intento che la descrive meglio espresso in una o due parole? |
| it-3 | Data la frase: [utterance]. Quale è il migliore intento corrispondente, se ce ne è uno, tra i seguenti: [potential intents]? |
| it-4 | [utterance] Scegli il miglior intente corrispondente basato sulla frase precedente. Opzioni: [potential intents] |
| it-5 | Estrarre l'intento in base a questa frase: [utterance]. |
| it-6 | "Dati i seguenti intenti: [potential intents]. Scegliere il più adatto, se presente, per questa frase: [utterance]." |
| de-1 | Betrachten wir diesen Satz: [utterance]. Welche Absicht beschreibt es am besten? |
| de-2 | Betrachten wir diesen Satz: [utterance]. Welche Absicht beschreibt sie am besten, ausgedrückt in ein oder zwei Worten? |
| de-3 | Angesichts der Phrase: [utterance]. Welche der folgenden Absichten passt am besten zu den folgenden: [potential intents]? |
| de-4 | [utterance] Wählen Sie die am besten passende Absicht basierend auf dem vorherigen Satz. Optionen: [potential intents] |
| de-5 | Extrahieren Sie die Absicht anhand dieses Satzes: [utterance]. |
| de-6 | "Angesichts der folgenden Absichten: [potential intents]. Wählen Sie, falls vorhanden, den am besten geeigneten Satz für diesen Satz aus: [utterance]." |

Table 8: A few examples of the incoming utterances and the respective predicted intents as predicted by the presented method and the the two main baselines. This is to facilitate the manual comparison of the generated intents of our method versus the baselines.

| Language | Utterance | Groundtruth | ZBERTA | GPT-3.5 | T0 |
|---|---|---|---|---|---|
| English | can you tell me where you get your exchange rates? | exchange-rates | exchange-rate | The intent of this utterance is to request information | exchange rates? get rates. idioms and expressions |
| English | can i choose a date for delivery? | card_delivery_estimate | delivery-date | The intent of this utterance is to inquire about the possibility of selecting a delivery date | choose date?..none of these choices |
| English | help! I can't find my card | lost_card | find-card | The intent for this sentence is to find a card that can't be found. | I have to find a card. |
| English | show how to recharge my card | recharging_card | recharge-card | How to recharge a card | How do you recharge a card? |
| Italian | mostrami come ricaricare con la mia carta | ricarica_carta | ricaricare-carta | Mostrare come ricaricare una carta. | Come puoi cricare la carta? |
| Italian | come vengono calcolati i tassi di cambio? | tassi_di_cambio | tasso-cambio | Comprendere come vengono calcolati i tassi di cambio. | vuoi sapere i tassi di cambio? |
| Italian | aiuto! non riesco a trovare la mia carta | carta_persa_o_rubata | trovare-carta | L'intento di questa richiesta è di trovare una carta | non riesco. i'm looking for my card. |
| Italian | com'è il tasso di cambio su questa app? | tassi_di_cambio | tasso-cambio | L'intento di questas richieste è di richiedere informazioni | è app. tasso di cambio |
| German | Ich muss Ihre Wechselkurse kennen | Wechselkurse | kennen wechselkurse, | Ich verstehe | dass Sie meine Wechselkurse kennen müssen. |
| German | Wann kann ich mit der Lieferung meiner neuen Karte rechnen? | Kartenlieferung | Karte erhalten | Es hängt von der Liefergeschwindigkeit ab | aber normalerweise dauert es 1 |
| German | Hilfe! Ich kann meine Karte nicht finden | verloren_Karte | Karte finden | Die Absicht dieses Satzes besteht darin, eine Karte zu finden, die nicht gefunden werden kann. | Ich muss eine Karte finden. |
| German | zeigen, wie ich meine Karte aufladen kann | aufladen_karte | aufladen-karte | wie ich eine Karte aufladen kann | wie ich eine Karte aufladen kann? |

Table 9: Exemplar transformations of Italian Banking77 dataset instances from intent classification to NLI. SD is the Synset definition extracted from WordNet which will be used to generate an augmented dataset from NLI having as a starting point any Text Classification dataset.

| Utterance | Extracted key-phrase | Wordnet Synset definition (SD) | Generated hypothesis |
|---|---|---|---|
| dove c'è supporto? | supporto | l'attività di mantenere o fornire attraverso la fornitura di necessità | questo testo riguarda **SD** |
| consegna della carta? | consegna | l'atto di consegnare o distribuire qualcosa | questo testo riguarda **SD** |
| ultimo pagamento? | pagamento | una somma di denaro pagato o una pretesa assolta | questo testo riguarda **SD** |

Table 10: Exemplar transformations of German Banking77 dataset instances from intent classification to NLI. SD is the Synset definition extracted from WordNet which will be used to generate an augmented dataset from NLI having as a starting point any Text Classification dataset.

| Utterance | Extracted key-phrase | Wordnet Synset definition (SD) | Generated hypothesis |
|---|---|---|---|
| Wo gibt es Unterstützung? | Unterstützung | die Tätigkeit des Unterhaltens oder Bereitstellens durch die Versorgung mit dem Nötigsten | In diesem Text geht es um **SD** |
| Kartenversand? | Lieferung | der Akt der Lieferung oder Verteilung von etwas | In diesem Text geht es um **SD** |
| letzte Zahlung? | Zahlung | ein gezahlter Geldbetrag oder eine befriedigte Forderung | In diesem Text geht es um **SD** |

Table 14: Five samples of new intent discovery through Zero-Shot-BERT-Adapters pipeline for English, Italian and German languages, full list available at `https://github.com/GT4SD/zero-shot-bert-adapters/blob/main/results/preds.csv`

| Ground-truth intent | Z-BERT-A-EN | Z-BERT-A-IT | Z-BERT-A-DE |
|---|---|---|---|
| exchange-rate | exchange-rate | cambio-soldi | geld-wechseln |
| card-delivery-estimate | delivery-time | consegna-data | pünktlich-liefern |
| lost-or-stolen-card | lost-card | carta-sparita | verlorene-karte |
| getting-virtual-card | virtual-card | carta-virtuale | virtuelle-karte |
| pin-blocked | pin-block | sbloccare-pin | entsperr-pin |

# B  Use-case for new emerging intents monitoring on real data

Here we show a practical use-case in which Zero-Shot-BERT-Adapters is successfully applied. From a series of users' telco assistance utterances in which its intent has not been recognized, we apply Zero-Shot-BERT-Adapters and we extract over a period of time the various emerging intents. These intents are then displayed in this dashboard we developed by applying also Semantic Similarity versus existing intents' categories in order to get a better glimpse of actual new emerging intents. We also compute various useful statistical information together with a graph showing an Hierarchical clustering using the Agglogmerative Clustering algorithm of the intents based on the semantic similarity score as clustering distance. The approach used in order to get the various intents grouped together is to first cluster similar extracted intents based on the cosine similarity of their embeddings, similarly to what we presented in the described work. Then assign to each cluster an intent name based on the most common occurrences of action and intent separately, taken from each cluster of intents. This helps normalize the same intent expressed thorugh a a set of the various possibile variants from different utterances, identifying a common intent among the ones with a similar semantic meaning. This approach is also cited in (Liu et al., 2021).

In the following Figure 4 we show a glimpse of the EID (Emerging Intent Dashboard) and its practical usage, backed by the Zero-Shot-BERT-Adapters pipeline.

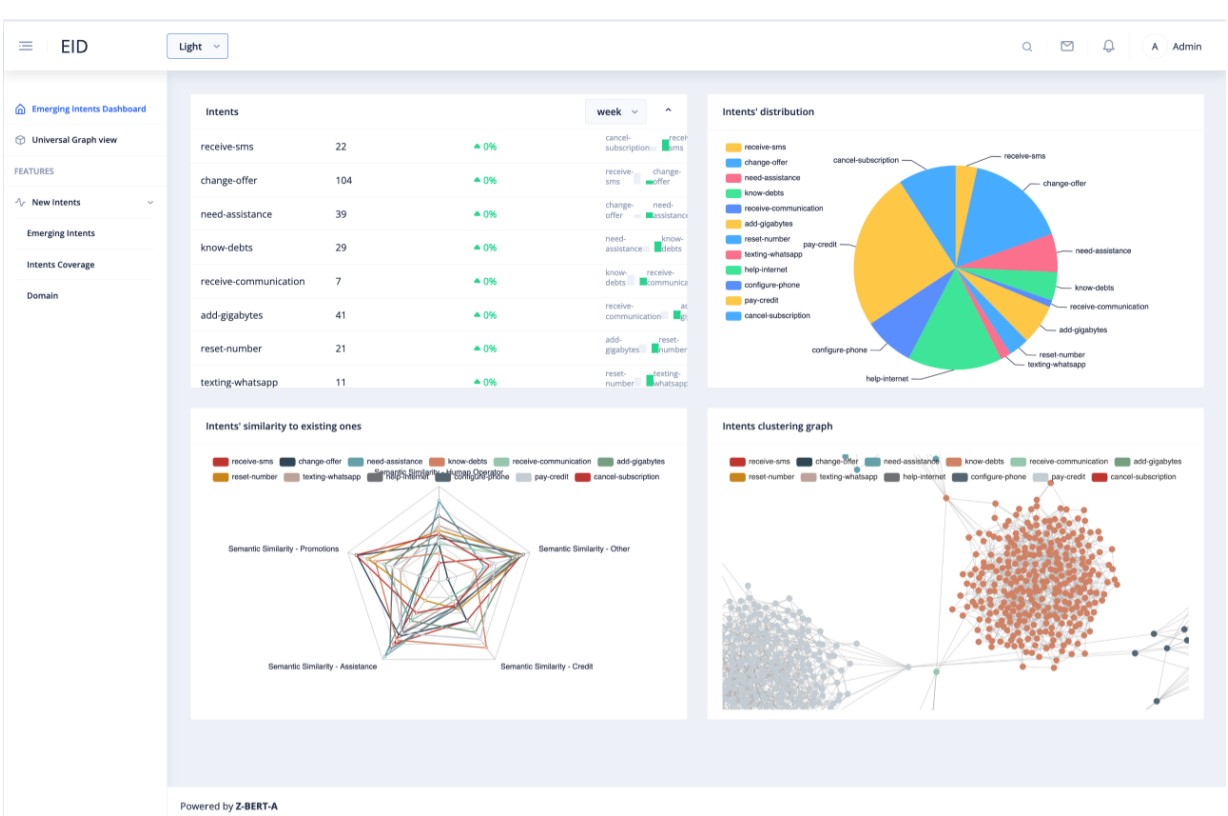

Figure 4: Emerging Intent Dashboard showing new emerging intent obtained from Zero-Shot-BERT-Adapters.