# OpenReview forum: "Zero-Shot-BERT-Adapters: a Zero-Shot Pipeline for Unknown Intent Detection"
_EMNLP/2023/Conference — EMNLP 2023 Findings_

### Official Review · Reviewer_wrS3 · 2023-07-31

**Soundness:** 3

**Excitement:**

3: Ambivalent: It has merits (e.g., it reports state-of-the-art results, the idea is nice), but there are key weaknesses (e.g., it describes incremental work), and it can significantly benefit from another round of revision. However, I won't object to accepting it if my co-reviewers champion it.

**Paper Topic And Main Contributions:**

This paper tackles the zero-shot intent detection tasks and proposes a two-stage zero-shot bert adapters (Z-BERT-A), which first leverages a dependency parser to extract a set of potential intents, then uses NLI methods relying on Bert models to classify on the candidate classes. Experimental results show this method can outperform a wide variety of baselines in both known intents zero-shot classification and unseen intent discovery.

**Questions For The Authors:**

In section 4, the intent generation process generates the candidates novel class for intent classification, but I wonder for a set of unseen classes, how to normalize same intention with different utterances, and how to determine the total number of new intent classes?

**Reasons To Accept:**

1. This paper focus on important tasks of both known intents zero-shot classification and unseen intent discovery, and can leverages dependency parsers to enhance the intent generation process.
2. Experimental results show the proposed methods are effective in zero-shot intent detection.


**Reasons To Reject:**

1. This work is better suited as a demo track paper, rather than a regular long paper.
2. The idea of using NLI to handle zero-shot learning tasks are quite common.


**Reproducibility:**

4: Could mostly reproduce the results, but there may be some variation because of sample variance or minor variations in their interpretation of the protocol or method.

**Reviewer Confidence:**

4: Quite sure. I tried to check the important points carefully. It's unlikely, though conceivable, that I missed something that should affect my ratings.

---

> ### Author Rebuttal · Authors · 2023-08-28
>
> We would like to thank the reviewer for the comments and for the provided review. Indeed, the use of NLI for zero-shot tasks is quite common, yet we propose to incorporate it in a simple yet effective pipeline which as highlighted also from the reviewer let us outperform much larger LLMs.
>
> The questions raised by reviewer regarding the normalization of identical intentions expressed through different utterance, is really interesting. The approach that we followed in the use case that we present in the Appendix is to first cluster similar extracted intents based on the cosine similarity of their embeddings. Then assign to each cluster an intent name based on the most common occurrences of action and intent separately. This is an interesting approach also cited (Liu et al., 2021) in one of the similar works of our paper. We are going to describe this step in more detail in the updated version of the paper even if this is not part of our contribution.

---

### Official Review · Reviewer_ihi6 · 2023-08-03

**Soundness:** 2

**Excitement:**

2: Mediocre: This paper makes marginal contributions (vs non-contemporaneous work), so I would rather not see it in the conference.

**Paper Topic And Main Contributions:**

The paper presents a technique for zero-shot intent classification. The authors make use of a BERT model finetuned on the NLI task and a dependency parser to discover new intents not seen before.

**Questions For The Authors:**

1. What is the frequency of examples in the dataset where the intent is explicitly mentioned in the sentence? If this is almost all of the cases, then my first reason to reject is not important. If there are a lot of examples without the intent mentioned, this method is fundamentally limited compared to LLMs which can generalize better than this approach (generate an intent without the intent being mentioned explicitly).

2. Are there any baselines that you could compare to for zero-shot intent classification? If so, why didn't you include them in Table 4?

3. What is the test set for Table 4?

**Reasons To Accept:**

Creativity: The authors present a creative pipeline that combines several components to predict new intents in a zero-shot setting.

Experiments: For many experiments, the authors show results for several different methods, comparing to a variety of LLMs.

**Reasons To Reject:**

Simplistic approach: The method presented in Algorithm 1 just extracts words from the sentence. If the intent word is not explicitly expressed in the sentence, this method will be incapable of generating the correct intent.

Lack of baseline in Table 4: The authors only present various settings for their model. I'm not familiar with this research area, so I have no idea if there are approaches in previously published work that outperform this method that the authors have left out.

Marginal improvement in Table 4: The difference in results for each approach are very small, so the benefit of the proposed method does not seem large.

Interpretability of remaining results: It's hard to compare the performance to the LLMs because they only use cosine distance. It's clear the model outperforms in semantic similarity (according to the semantic encoder models used), but for more trustworthy results, a small sample of human evaluations should be used as well to be sure that this method outperforms the LLMs in the zero-shot setting. Another option would be to modify the LLM experiment such that label F1 scores could be produced (use a verbalizer to map LLM output to intent classes).

**Reproducibility:**

3: Could reproduce the results with some difficulty. The settings of parameters are underspecified or subjectively determined; the training/evaluation data are not widely available.

**Reviewer Confidence:**

2: Willing to defend my evaluation, but it is fairly likely that I missed some details, didn't understand some central points, or can't be sure about the novelty of the work.

---

> ### Author Rebuttal · Authors · 2023-08-28
>
> We would like to thank the reviewer. We appreciate the comments and the suggestions to improve our paper.
>
> Regarding Q1, it is indeed worth noting that in all instances within the dataset we employed, the intended intent is explicitly articulated within the provided utterances. Notably, since the dataset comprises real-world utterances concerning bank transactions, it becomes evident that this pattern holds true for standard dialogues of this nature. This inherent characteristic significantly influenced the development of our pipeline. Furthermore, based on the results indeed this fact helps us to perform better than much larger LLMs. We will try to emphasize further on this aspect in the updated version of the paper.
>
> Regarding Q2, we can consider all the prompts that have been used for LLMs which include the possible intents in them as further zero-shot baselines for this task. These prompts are prompt en-3, en-4 and en-6 present in the Table VIII of the Appendix. Based on Table VI and Table VII, our method outperforms all these baselines in terms of cosine similarity for this task. Furthermore, in the newly added from us German translated version of the dataset (see review cqfr) our model is able to outperform these baselines also for the German language. In the updated version, we are going to consolidate all these results to highlight better this comparison. Furthermore, as reviewer raised an interesting point regarding other ways of evaluating the performance of our work. We attach below a few examples of the incoming utterances and the respective predicted intents as predicted by our method and the rest baselines. A completed set of such instances will be added in the appendix of the paper to facilitate the manual comparison of the generated intents of our method versus the baselines. Based on this manual investigation, we observe that our model is generally slightly better in English, while for the other language its performance is significantly better than the baselines. This helps to have concise intent answers while being able to generalize over completely unseen intent classes for each different request.
>
> Here some examples:
>
> English:
>
> Utterance: can you tell me where you get your exchange rates?, Groundtruth: exchange-rates, Zberta: exchange-rate, Openai: The intent of this utterance is to request information., T0: exchange rates? get rates. idioms and expressions
>
> Utterance: can i choose a date for delivery?, Groundtruth: card_delivery_estimate, Zberta: delivery-date Openai: The intent of this utterance is to inquire about the possibility of selecting a delivery date. T0: choose date?..none of these choices
>
> Italian:
>
> Utterance: aiuto! non riesco a trovare la mia carta, Groundtruth: carta_persa_o_rubata, zberta: trovare-carta, Openai: L’intento di questa richiesta è di trovare una carta, T0: non riesco. i’m looking for my card.
>
> Utterance: com’è il tasso di cambio su questa app?, Groundtruth: tassi_di_cambio, Zberta: tasso-cambio openai: L’intento di questas richieste è di richiedere informazioni. T0: è app. tasso di cambio
>
> German:
>
> Utterance: Ich muss Ihre Wechselkurse kennen, Groundtruth: Wechselkurse, Zberta: kennen wechselkurse, Openai: Ich verstehe, dass Sie meine Wechselkurse kennen müssen.
>
> Utterance: Wann kann ich mit der Lieferung meiner neuen Karte rechnen? Groundtruth: Kartenlieferung, Zberta: Karte erhalten, Openai: Es hängt von der Liefergeschwindigkeit ab, aber normalerweise dauert es 1
>
> The test set for Table 4 and all the rest evaluations and tables presented in this work is the provided standard test set from BANKING-77 and SNLI. We are going to mention this clearly in the revised version of the paper.

---

### Official Review · Reviewer_cqfr · 2023-08-03

**Soundness:** 2

**Excitement:**

3: Ambivalent: It has merits (e.g., it reports state-of-the-art results, the idea is nice), but there are key weaknesses (e.g., it describes incremental work), and it can significantly benefit from another round of revision. However, I won't object to accepting it if my co-reviewers champion it.

**Paper Topic And Main Contributions:**

This paper proposed a method to do zero-shot intent classification, it can be applied to BERT-based transformer models. The method contains two stages, where for stage-1, the dependency parser is used to get potential intents and in stage-2 the zero-shot classification is performed for final output. Experiments are done on public datasets to verify the effectiveness of the proposed method.

**Questions For The Authors:**

1. How does this method compare with other existing language model adapters.
2. What are the tunable parameters and what are the frozen parameters in the model?
3. What is the size of the trainable parameters in the proposed method?
4. The model has been used in English and Italian, can experiments be added to  one more language to better prove the multilingual ability?


**Reasons To Accept:**

The paper designed a method as a BERT adapter to handle the zero-shot intent discovery task. The model has been evaluated on two datasets and achieved state-of-the-art performance.

**Reasons To Reject:**

The contribution of the paper is not very clear, how does this method compare with other existing language model adapters.
More ablation study could be done to prove the effectiveness of components in the model architecture.


**Reproducibility:**

5: Could easily reproduce the results.

**Reviewer Confidence:**

3: Pretty sure, but there's a chance I missed something. Although I have a good feel for this area in general, I did not carefully check the paper's details, e.g., the math, experimental design, or novelty.

**Typos Grammar Style And Presentation Improvements:**

Here are some minor notes that the author may consider:
1. In line-184, building a pipeline [that is] able to handle unseen classes.
2. As the proposed method is a two-stage pipeline, it could be better if stage-1 and stage-2 can be clearly illustrated in the pipeline figure.

---

> ### Author Rebuttal · Authors · 2023-08-28
>
> We would like to thank the reviewer for the comments and the suggestions to improve our paper.
>
> The primary focus of this study is centered around the proposed two-step pipeline. The first step involves identifying potential intents within a given utterance, leveraging syntax dependencies. Subsequently, in the second step, an evaluation is conducted to determine which of these suggestions accurately encapsulates the true intent of the utterance. This straightforward yet highly effective pipeline holds dual significance:
>
> (i) It can be readily employed as-is, obviating the need for predefined intent categories. This capability facilitates the extraction of intents from incoming utterances.
>
> (ii) It can be seamlessly integrated into other intent detection pipelines, providing support in identifying novel and unfamiliar intents that emerge from incoming utterances.
>
> The incorporation of adapters (Q1) serves primarily to reduce the memory footprint of our pipeline and to expedite the training time of the corresponding model. In our research, we have utilized one of the most widely recognized adapter architectures (Pfeiffer  et al., 2020). However, it is important to acknowledge that further refinements in this aspect could be possible.
>
> Consequently, our experimental endeavors have been concentrated exclusively on the intent detection task. This approach has been motivated by our main objective of proposing an unknown intent identification methodology.
>
> Regarding the Q2 and Q3 and as we highlight in the paper we kept the whole BERT model frozen and only update the added Adapter layers to minimize training time and memory footprint. This means that our final model has almost 111 million parameters (110 million parameters BERT + 896'066 parameters adapters) from which only 896066 are updated during training. We will revise the respective part of the paper to illustrate further these details.
>
> To prove the multilingual ability as it is suggested in Q4, we performed further experiments using a translated version of Banking dataset in German. Aligned, with performance in English and Italian our model outperforms (cosine similarity scores - ZBERTA: 0.36, GPT-3.5 without intents candidates: 0.05, GPT-3.5 with intent candidates: 0.17). We are going to update the respective section of the paper to include these further evaluations.

---

### Meta-Review · Area_Chair_uyvA · 2023-09-16

**Recommendation:** 1

**Metareview:**

The paper presents a method for zero-shot intent discovery using a BERT adapter, which is evaluated on two datasets and achieves improved performance over compared baselines. However, one reviewer complains that the contribution of the paper is not clearly stated and its effectiveness compared to other language model adapters is unclear. Another reviewer thought more baselines and comparison should be made, while the differences in results between the proposed method and other approaches are marginal. The authors are asked to provide more information on how their method compares to other existing language model adapters, the tunable and frozen parameters in the model, the size of the trainable parameters, and whether there are any baselines that could be compared to for zero-shot intent classification. The work was also considered more suitable for a demo track paper instead of a regular long paper by one reviewer.

---

### Decision · Program_Chairs · 2023-10-07

**Decision:**

Accept-Findings

**Comment:**

The paper presents a method for zero-shot intent discovery using a BERT adapter, which is evaluated on two datasets and achieves improved performance over compared baselines. However, one reviewer complains that the contribution of the paper is not clearly stated and its effectiveness compared to other language model adapters is unclear. Another reviewer thought more baselines and comparison should be made, while the differences in results between the proposed method and other approaches are marginal. The authors are asked to provide more information on how their method compares to other existing language model adapters, the tunable and frozen parameters in the model, the size of the trainable parameters, and whether there are any baselines that could be compared to for zero-shot intent classification. The work was also considered more suitable for a demo track paper instead of a regular long paper by one reviewer.